# Combining Three Tyrosine Kinase Inhibitors: Drug Monitoring Is the Key

**DOI:** 10.3390/ijms24065518

**Published:** 2023-03-14

**Authors:** Quentin Dominique Thomas, Nelly Firmin, Litaty Mbatchi, Alexandre Evrard, Xavier Quantin, Fanny Leenhardt

**Affiliations:** 1Montpellier Cancer Institute (ICM), 34090 Montpellier, France; 2Montpellier Cancer Research Institute (IRCM), University of Montpellier (UM), 34090 Montpellier, France; 3Pharmacokinetics Laboratory, Faculty of Pharmacy, University of Montpellier, 34090 Montpellier, France

**Keywords:** lung cancer, gastrointestinal stromal tumor (GIST), osimertinib, crizotinib, imatinib, tyrosine kinase inhibitor, therapeutic drug monitoring

## Abstract

A combination of tyrosine kinase inhibitors (TKIs) is likely to be a therapeutic option for numerous oncological situations due to high frequency of oncogenic addiction and progress in precision oncology. Non-small cell lung cancer (NSCLC) represents a subtype of tumors for which oncogenic drivers are frequently involved. To the best of our knowledge, we report the first case of a patient treated with three different TKIs. Osimertinib and crizotinib were administered concurrently for an epidermal growth factor receptor (*EGFR*)-mutated NSCLC developing a *MET* amplification as a resistance mechanism to osimertinib. Simultaneously, imatinib was administered for a metastatic gastrointestinal stromal tumor. The progression-free survival was 7 months for both tumors with this tritherapy. The use of therapeutic drug monitoring to assess plasma concentrations of each TKI was a powerful tool to manage the toxicity profile of this combination (creatine phosphokinase elevation) while preserving an optimal exposure to each TKI and treatment efficacy. We observed an imatinib over-exposition related to crizotinib introduction, probably explained by drug–drug interaction mediated by crizotinib enzymatic inhibition on cytochrome P-450 3A4. Posology adjustment due to therapeutic drug monitoring was probably involved in the good survival outcome of the patient. This tool should be used more routinely for patients treated by TKIs to prevent co-treatment interactions and, in particular, for patients receiving TKI combinations to obtain optimal therapeutic exposure and efficacy while reducing possible side-effects.

## 1. Introduction

Abnormal activation of tyrosine kinases due to mutations, translocation, or amplification is involved in tumorogenesis, cancer cell proliferation, and metastasis [1]. As such, tyrosine kinases have emerged as major targets for drug discovery. Tyrosine kinase inhibitors (TKIs) have played an important role in improving the survival of patients with tumors. Imatinib was the first TKI to receive FDA approval in 2001 for the treatment of chronic myelogenous leukemia. It has subsequently been extended for the treatment of malignant metastatic and/or unresectable gastrointestinal stromal tumors (GIST) [2]. Multiple targets have emerged over the last twenty years contributing to the significant progress in cancer treatment [3]. Numerous driver mutations have been identified in non-small cell lung cancer (NSCLC) and are targetable by TKIs [4]. Epidermal growth factor receptor (EGFR) activating mutations occur in 10–15% of patients with NSCLC. Osimertinib, a third-generation EGFR TKI, is currently recommended in first metastatic line for patients with an *EGFR*-mutated (deletion exon 19 or L858R mutation in exon 21) NSCLC [5]. *MET* amplification represents the main mechanism of acquired resistance to osimertinib (10–15% of cases) [6,7]. Clinical trials are currently underway to find effective strategies in this setting (NCT05015608/NCT03940703). Several drug associations of anti-EGFR and anti-MET show promising results for these patients [8,9]. Unfortunately, history of a second active cancer within 5 years of screening is an exclusion criteria for almost all clinical trials. For patients with a second cancer, crizotinib, an FDA-approved TKI in patients treated for an NSCLC with *ALK* or *ROS-1* translocation, is also active against *MET*-amplification. It is an off-label treatment option for patients not suitable for clinical trials developing a *MET* amplification under osimertinib [10,11]. TKI combination allows for bypassing resistance mechanisms but exposes the patient to major risk of toxicity or drug–drug interactions (between TKIs or co-treatment). In addition, the dose of TKIs needs to be adjusted frequently in case of adverse events. Therapeutic drug monitoring (TDM) is a biological monitoring tool to compare plasma concentration of each TKI with international target concentration guidelines in order to enhance the efficacy and safety of these combinations. Dose adjustment only on toxicity occurrence is no longer relevant in oncology. TKI dosage can be optimized with TDM to avoid under/over-exposition of TKI in a prospective way [12]. We report here the case of a patient with two active cancers: a GIST treated with imatinib and an *EGFR*-mutated NSCLC that developed a *MET* amplification during osimertinib treatment. We used a combination of three TKIs (imatinib, osimertinib, and crizotinib) to treat both diseases. We report here the use of therapeutic drug monitoring to adapt the dose of each TKI in order to limit the risk of toxicity while ensuring optimal anti-tumor efficacy.

## 2. Case Report

### 2.1. Initial Management of GIST

The patient is a 68-year-old woman with a main medical history of congenital unique kidney and high blood pressure without chronic medication and no family history of cancer. She was diagnosed with an abdominal tumor mass in March 2002 and treated with surgery. The initial diagnosis was a leiomyosarcoma developed on ileum. Pelvic relapse occurred one year after. The biopsy analysis by anatomopathologists experts in mesenchymal tumors resulted in a diagnosis of GIST. Immunohistochemistry showed C-Kit and vimentin positive expression; CD-34, AML, desmine, and H-Caldesmone were negative. Given the diagnosis of GIST and given the locally advanced disease with a very large pelvic mass, the patient was treated by imatinib at 400 mg per day in the French sarcoma group study BFR14 [13]. The BFR14 study was a study in which we evaluated the interruption and the rechallenge of imatinib at 1, 3, and 5 years in advanced GIST. The patient was randomized into the 5-year imatinib group. The treatment was carried out until April 2008. She had pelvic progression after a 7-month-free interval and the imatinib was rechallenged at the same dose. In 2014, she presented with an abdominal parietal oligoprogression that was surgically managed. Genotyping revealed a mutation in exon 11 of *KIT* (c.1756T > A, p.W557R). In 2018, hepatic oligoprogression occurred and liver biopsy confirmed GIST metastasis and identified an exon 13 mutation of *KIT* associated with the known exon 11 *KIT* mutation. This hepatic oligoprogression was treated with radiofrequency.

### 2.2. Diagnosis of EGFR Lung Adenocarcinoma: Concomitant Administration of Imatinib + Osimertinib

In 2020, the patient developed pulmonary progression with multiple bilateral lesions observed on computed tomography (CT)-scan. All lung lesions were millimetric, suggestive of metastatic miliary disease, except for one upper-right lobe lesion that measured 31 mm in major axis. A trans-thoracic biopsy of the main lung lesion identified a lung adenocarcinoma (LUAD), TTF1+ on immunohistochemistry. A positron emission tomography (TEP) scan showed diffuse mediastinal lymph node involvement confirmed histologically by bronchial echo-endoscopy showing lymph node metastases from TTF1+ LUAD. Genotyping realized on lung tissue revealed a deletion of exon 19 of *EGFR*. Given the unresectable status and the *EGFR*-mutated LUAD, systemic treatment with afatinib, a second-generation EGFR TKI, was initiated in combination with imatinib. Safety was impacted by the occurrence of grade 3 diarrhea (CTCAE v5.0). Afatinib was switched for osimertinib at 80 mg/day in September 2020 as the safety profile of third-generation EGFR TKI is known to be better than first- or second-generation [14]. At that time, imatinib treatment was suspended due to the stability of the GIST for several years and the poor prognosis of LUAD. This therapeutic strategy allowed a response on all pulmonary and mediastinal lesions but the known lesion of the right upper lobe rapidly progressed, justifying a stereotactic radiotherapy delivering 60 gy in four fractions from 2 February 2021 to 16 February 2021. In April 2021, a liver progression of the GIST warranted reintroduction of imatinib in combination with osimertinib. In this TKI combination used to treat both locations, imatinib was reintroduce at a reduced dose (200 mg/day). After two weeks of combination therapy, the patient developed grade 3 creatine phosphokinase (CPK) elevation (CTCAE v5.0) with acute renal failure, warranting dose reduction of osimertinib to 40 mg/day. This dose adaptation allowed the correction of side effects and the dose of imatinib was increased to 300 mg at follow-up.

TDM is a clinical-biological tool for kinase inhibitors to optimize dosages, maintaining safety and efficacy for drugs with narrow therapeutic range. Imatinib is a molecule of choice according to extensive pharmacokinetic/pharmacodynamic (PK/PD) correlation data available. In GIST, a trough imatinib concentration of approximately 1000 ng/mL is the therapeutic goal for efficacy [15]. For osimertinib, the exploratory target of 166 ng/mL is based on the observed mean trough level, but recent data suggest that high concentrations of osimertinib (>235 ng/mL) are negatively correlated with survival [16].

A first plasma monitoring of osimertinib and imatinib concentration was performed in May 2021. The trough concentration (C_trough_) of osimertinib was 275 ng/mL at a dose of 40 mg/day and C_trough_ of imatinib was 1198 ng/mL at 300 mg/day. Even at adapted doses, the patient seemed to be correctly exposed to these two treatments, allowing us to continue at the same doses.

### 2.3. Appearance of MET Amplification as Resistance Mechanism to Osimertinib: Concomitant Administration of Imatinib + Osimertinib + Crizotinib

In July 2021, after 10 months of osimertinib, the CT-scan showed stability of the lung and liver lesions with partial response of the peritoneal carcinosis but for the appearance of a 22 mm left adrenal mass syndrome. An adrenal biopsy was performed in October 2021, showing a metastasis of the LUAD. Genotyping on this biopsy revealed the *EGFR* mutation, and *MET* amplification was identified as a mechanism of resistance to osimertinib. Crizotinib was introduced in off-label combination with osimertinib to treat this progression. Crizotinib was introduced at a reduced dose of 250 mg/day to assess the tolerability of the combination therapy; in addition, imatinib treatment was initially suspended. One month later, TDM highlighted good plasma concentration to osimertinib 40 mg, combined with crizotinib 250 mg, with C_trough_ above observed value (≥235 ng/mL for crizotinib) [17]. Safety was characterized by the occurrence of grade 3 cytolysis (CTCAE v5.0) with alanine aminotransferase increased at 5.2N, requiring a decrease in the dose of crizotinib to 200 mg/day, allowing a normalization of transaminases. Although the dose reduction resulted in a limited plasma concentration of crizotinib (131 ng/mL), the combination was efficient. This strategy provided a partial response in the left adrenal lesion from 72 × 32 mm to 54 × 18 mm after 3 months of bitherapy. However, the progression of the liver lesion and subdiaphragmatic adenopathy was indicative of GIST progression. In this context of two active cancers, a tritherapy was performed with reintroduction of imatinib 200 mg/day in January 2022. In order to evaluate plasma concentration of the tritherapy, TDM of crizotinib and osimertinib was performed at baseline, indicating a suboptimal C_trough_ for both TKIs (C_trough_ crizotinib 200 mg/day: 131 ng/mL; C_trough_ osimertinib 40 mg/day: 141 ng/mL). Two weeks after adding imatinib, C_trough_ of imatinib was more than 2400 ng/mL, well above the expected threshold of 1100 ng/mL. The overall tolerance being correct, treatments were maintained at the same dosage. Over-exposition of imatinib plasma concentration was confirmed with a new assay 2 months later, indicating a C_trough_ of 3250 ng/mL. CT-scan performed in March 2022, after 3 months of tritherapy, indicated a stabilization of the LUAD and the GIST. Considering the results of CT-scan and the TDM, indicating an over-exposure to imatinib, a dosage adjustment was made by decreasing the imatinib dosage to 100mg/day and increasing the crizotinib dosage to 250 mg/day, in view of the limited C_trough_ in January 2022. One month after, TDM was performed for each TKI, highlighting optimal C_trough_ for crizotinib (250 mg/d) and osimertinib (40 mg/d) but under-exposition for imatinib (100 mg/d). In June 2022, the patient developed grade 2 CPK elevation (CTCAE v5.0) related to crizotinib. CT-scan showed a hepatic progression of the GIST, justifying the suspension of crizotinib and the increase of imatinib to 200 mg/d. In August 2022, after 7 months of tritherapy, the patient presented a hepatic progression of the GIST and a left adrenal progression of the LUAD, requiring therapeutic modification to chemotherapy. (Figure 1 and Figure 2).

## 3. Discussion

To the best of our knowledge, we present one of the only cases of a patient treated simultaneously by three tyrosine kinase inhibitors for two different cancers. The patient had a progression-free survival (PFS) of 7 months with osimertinib and crizotinib for a metastatic *EGFR*-mutated LUAD presenting a *MET* amplification as acquired resistance to EGFR-TKI associated with imatinib administered for a metastatic GIST. Progression after osimertinib for *EGFR*-mutated NSCLC is unavoidable. Better understanding of the varied molecular mechanisms of resistance to this TKI offer several opportunities for the use of targeted agents after osimertinib failure. A major ongoing focus of clinical research is the targeting of *MET* signaling pathway with current developments of bispecific EGFR-MET antibody or TKI combination targeting both pathways [18]. Wang et al. have reported on a series of 14 patients treated with crizotinib with or without EGFR-TKI for *EGFR*-mutated LUAD with acquired *MET* amplification after failure of EGFR-TKI therapy. Similar results were obtained, with a disease control rate of 85.7% (12/14 patients) and a median PFS of 12.6 months for patients treated by crizotinib plus EGFR-TKI [19]. The main severe adverse event for patients treated with combination was aminotransferase rise. One of the main challenges faced when combining multiple TKIs is the management of toxicity induced by direct interactions between these molecules but also with patient co-treatments. Indeed, these TKIs may cause pharmacokinetic drug–drug interactions because most of them are substrates of intestinal and/or hepatic cytochrome P-450 (CYP) enzymes (mostly CYP3A4) and membrane transporters [20]. Clinico-biological evaluation allows for the monitoring of TKI tolerance and an adaptation of the dosage of each TKI is often necessary to ensure the simultaneous use of the molecules. However, the impact of dose adjustments on treatment efficacy is often underestimated. Thus, the use of TDM allows for characterization of plasma concentration of treatments and for dose adjustment in personalized way, as is already described in a large number of treatments in oncology and other disciplines. TDM enables the characterization of pharmacokinetic (PK) variability sources as drug–drug interactions. It should be noted that the patient was not taking any other medication. In this case report, the main hypothesis retained to explain the over-exposure to imatinib is the phenomenon of enzymatic inhibition due to the association with crizotinib primarily and osimertinib secondarily. Thus, it was observed that even combined with osimertinib and at a higherdose (300 mg/d, May 2021), C_trough_ of imatinib was correct. When crizotinib was added (November 2021) imatinib was overdosed, even though it was 200 mg/day. The main hypothesis retained to explain this drug–drug interaction mediated by crizotinib is an enzymatic inhibition on CYP3A4 mainly. During crizotinib clinical development, co-administration with Midazolam increases the total exposition (AUC) values for Midazolam by 265% [21]. Plasma exposure of osimertinib was not significantly impacted by the other TKIs due to the dosage adjustments of each (imatinib increased while crizotinib decreased and in the other direction as well). PK-guided dose optimization of these three oral targeted therapies reduces over- and under-exposure. This clinical case highlights that the standard dosages for most of these oral targeted therapies must be moderated against the actual plasma concentration of the patients. In this case report, although TKI dose was reduced by half, the patient benefited from three TKIs over several months, allowing an alternating clinical response and, above all, correct clinical and biological safety. The use of TDM would be even more refined if it were performed more frequently or if global exposure (AUC) was estimated using a better pharmacokinetic biomarker. Small-molecule inhibitors, mainly represented by TKIs, have become increasingly widely used in the last few years due to advances in precision medicine and theranostic tools in oncology [22]. Combination TKIs are likely to be a treatment option for many types of cancers in the future. In addition, the improvement in tumor control due to the progress of targeted therapies leads to an increase in the longevity of patients who are more likely to develop a second cancer. This type of situation requiring the combination of several TKIs is likely to be encountered more frequently in the years to come [23]. The main objective will be to maintain optimal anti-tumor activity while ensuring acceptable tolerance. The use of TDM appears to be an interesting tool in this objective.

## Figures and Tables

**Figure 1 ijms-24-05518-f001:**
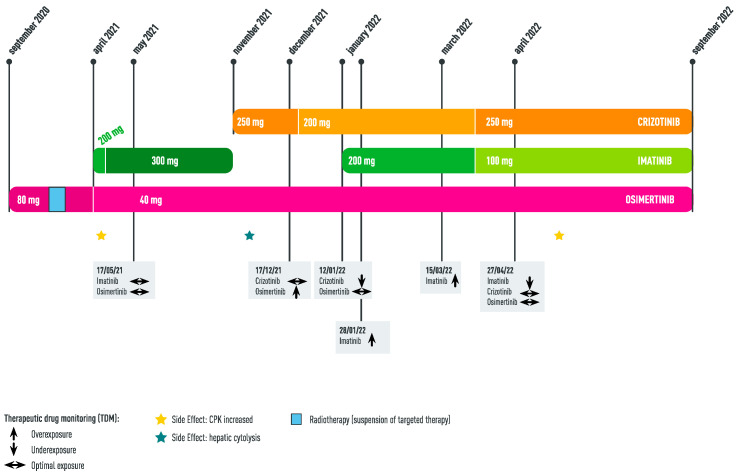
Timeline of tyrosine kinase inhibitor combination and associated side effects.

**Figure 2 ijms-24-05518-f002:**
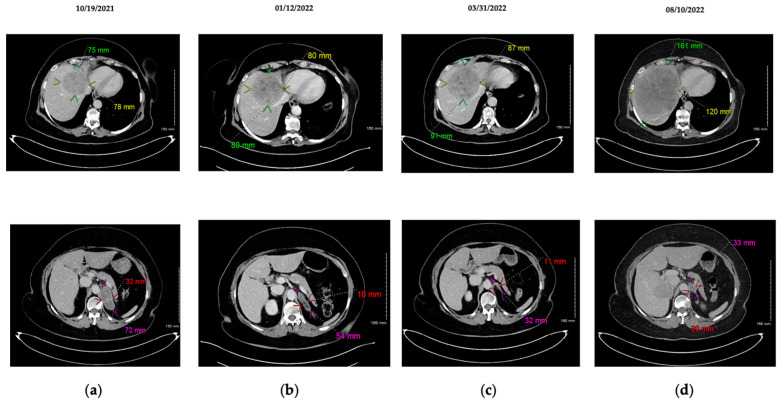
Hepatic (**top**) and adrenal (**bottom**) CT-scan illustrating tumoral evolution with combination of tyrosine kinase inhibitors. (**a**) Baseline with osimertinib + crizotinib combination; (**b**) After 3 months with osimertinib + crizotinib; adrenal partial response; non-significant trend in hepatic progression; introduction of imatinib; (**c**) Stable disease after 3 months of tritherapy (osimertinib + crizotinib + imatinib); non-significant trend in hepatic progression; (**d**) Hepatic and adrenal progression after 7 months of tritherapy.

## Data Availability

No new data were created. Data are available in patients medical records.

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
