# Peer review of "Combining Three Tyrosine Kinase Inhibitors: Drug Monitoring Is the Key"

_ijms, 2023, doi:10.3390/ijms24065518_

Round 1

Reviewer 1 Report

Tyrosine kinase inhibitors (TKIs) represent the standard treatment for patients with non-small cell lung cancer (NSCLC) harboring epidermal growth factor receptor (EGFR) mutations. The duration of the response is, however, limited in time owing to the development of resistance mechanisms to both first- and second-generation agents such as MET oncogene amplification. This case report describes 

the first case of a  patient treated with three different TKIs. Osimertinib and Crizotinib were administered concurrently for an Epidermal growth factor receptor (EGFR) mutated NSCLC developing a MET amplification as resistance mechanism to Osimertinib (EGFR-TKI). Simultaneously, Imatinib was administered for a metastatic gastrointestinal stromal tumor. The progression-free survival was 7 months for both tumors with this tritherapy. The authors found  an Imatinib over-exposition related to Crizotinib introduction probably ex-plained by drug-drug interaction mediated by Crizotinib enzymatic inhibition on CYP3A4. Posology adjustment thanks to the therapeutic drug monitoring were probably involved in the good survival outcome of the patient. This tool should be used more routinely for patient treated by TKIs to prevent co-treatment interactions and even more for patients receiving TKIs combination to obtain 

optimal therapeutic exposure and efficacy while reducing possible side-effect. This case report is very interesting 

Author Response

We would like to thank Reviewer 1 for taking the time to evaluate our article. We are also pleased that Reviewer 1 found the current version satisfactory without requesting any changes to the content.

Reviewer 2 Report

This case study demonstrates the potential effectiveness of using several targeted drugs in the treatment of cancer. It highlights how crucial TDM is for controlling pharmacological side effects, adjusting dosing schedules, and assessing the efficacy of treatments. The long-term efficacy and safety of these drug combinations need to be determined through further research, but this case study provides a critical example of how personalized targeted therapy approaches may be used to treat cancer.

Since they point out that TDM can be an important tool for assessing the efficacy of treatments, adjusting dosing regimens, and managing drug-drug interactions, the authors should emphasize the methodology they used and include a methods explanation of TDM. 

It would also be preferable to divide the “2. Case report” section into multiple subsections. This would provide a more organized and comprehensive overview of the case study, making the results easier to comprehend.

Authors mentioned PK for the first time in the 189. line: "TDM enables to characterize PK variability sources as drug-drug interaction." However, they did not define PK. 

Author Response

We would like to thank reviewer 2 for his proofreading and helpful comments on our article. Here is a point-by-point response to the comments :

1-Since they point out that TDM can be an important tool for assessing the efficacy of treatments, adjusting dosing regimens, and managing drug-drug interactions, the authors should emphasize the methodology they used and include a methods explanation of TDM. 

--> We have included in our article several paragraphs specifying the known and expected plasma exposure data for each of the molecules :

For imatinib and osimertinib lines 118-124
For crizotinib lines 143-144

2- It would also be preferable to divide the “2. Case report” section into multiple subsections. This would provide a more organized and comprehensive overview of the case study, making the results easier to comprehend.

We fully agree with reviewer 2 and have separated the "Case report" section into 3 subdivisions

3-Authors mentioned PK for the first time in the 189. line: "TDM enables to characterize PK variability sources as drug-drug interaction." However, they did not define PK. 

--> We have amended this error line 221